# Transcript Variants of Genes Involved in Neurodegeneration Are Differentially Regulated by the APOE and MAPT Haplotypes

**DOI:** 10.3390/genes12030423

**Published:** 2021-03-15

**Authors:** Sulev Koks, Abigail L. Pfaff, Vivien J. Bubb, John P. Quinn

**Affiliations:** 1Perron Institute for Neurological and Translational Science, Perth, WA 6009, Australia; Abigail.Pfaff@murdoch.edu.au; 2Centre for Molecular Medicine and Innovative Therapeutics, Murdoch University, Perth, WA 6150, Australia; 3Department of Pharmacology and Therapeutics, Institute of Systems, Molecular and Integrative Biology, University of Liverpool, Liverpool L69 3BX, UK; jillbubb@liverpool.ac.uk (V.J.B.); jquinn@liverpool.ac.uk (J.P.Q.)

**Keywords:** Apolipoprotein E (ApoE), microtubule associated protein (MAPT), synuclein alpha (SNCA), eQTL, TOMM40, KANSL1, mitochondria, Parkinson’s Progression Markers Initiative (PPMI)

## Abstract

Genetic variations at the Apolipoprotein E (ApoE) and microtubule-associated protein tau (MAPT) loci have been implicated in multiple neurogenerative diseases, but their exact molecular mechanisms are unclear. In this study, we performed transcript level linear modelling using the blood whole transcriptome data and genotypes of the 570 subjects in the Parkinson’s Progression Markers Initiative (PPMI) cohort. ApoE, MAPT haplotypes and two SNPs at the SNCA locus (rs356181, rs3910105) were used to detect expression quantitative trait loci eQTLs associated with the transcriptome and differential usage of transcript isoforms. As a result, we identified 151 genes associated with the genotypic variations, 29 *cis* and 122 *trans* eQTL positions. Profound effect with genome-wide significance of ApoE e4 haplotype on the expression of TOMM40 transcripts was identified. This finding potentially explains in part the frequently established genetic association with the APOE e4 haplotypes in neurodegenerative diseases. Moreover, MAPT haplotypes had significant differential impact on 23 transcripts from the 17q21.31 and 17q24.1 loci. MAPT haplotypes had also the largest up-regulating (256) and the largest down-regulating (−178) effect sizes measured as β values on two different transcripts from the same gene (LRRC37A2). Intronic SNP in the SNCA gene, rs3910105, differentially induced expression of three SNCA isoforms. In conclusion, this study established clear association between well-known haplotypic variance and transcript specific regulation in the blood. APOE e4 and MAPT H1/H2 haplotypic variants are associated with the expression of several genes related to the neurodegeneration.

## 1. Introduction

Apolipoprotein E (ApoE) epsilon 4 haplotype (e4) is the major risk factor for the Alzheimer disease (AD) [1]. Three main haplotype versions, e4, e3 and e2, exist for ApoE and these are defined by the single nucleotide polymorphisms (SNPs) rs429358 and rs7412. The effect of e4 is dosage-dependent on the risk of AD and at the same time e2 variant of ApoE exhibits protective effect for late onset AD [2]. This protective effect has been verified in the animal models overexpressing e4 or e3 variants of human ApoE genes [3]. The increased risk for AD with ApoE e4 is also evident in diverse populations and it is considered that all the genetic risk for AD in this region is accounted for by the ApoE e4 haplotype [4]. In addition, ApoE variants are not only involved in AD, but several studies also report an association between ApoE e4 and Parkinson’s Disease (PD) [5]. The association between ApoE and PD is complex and seems to be population specific to certain PD subtypes or specific symptoms [5,6,7]. It is most commonly associated with cognitive decline in PD or Lewy body dementia [6,7]. A recent detailed analysis on three longitudinal cohorts identified ApoE e4 to be associated with cognitive progression of PD [8]. Despite decades of research, the molecular reasons for the ApoE e4 association are not yet clear. Genetic variations in the TOMM40 gene have been described to modify the effects of ApoE variants due to linkage disequilibrium (LD) [9]. Indeed, several subsequent studies confirmed that ApoE and TOMM40 have a combined effect on the heritability of AD and analysis of both genes improves the precision to estimate the age of onset of AD [9,10,11,12]. While there is a strong genetic link between TOMM40 and ApoE the precise molecular interaction between ApoE and TOMM40 is not understood. The strong functional association could also be exhibited on their expression as the adjacent and close to one another on human chromosome 19 suggestive of overlapping transcriptional regulation.

Microtubule-associated protein tau (MAPT) is considered a significant risk factor for neurodegenerative diseases, including PD [13,14,15,16,17,18,19]. Tau protein is biochemically well-characterised and its role in aggregation and neurodegeneration was well established [18]. Recent genome wide association studies (GWAS) have repeatedly identified the MAPT locus as a significant independent risk locus for PD, confirming early genetic association studies indicating the link between MAPT and PD [20,21,22]. Genetic variation of the MAPT locus is described by the H1 and H2 haplotypes, that can be identified by different SNPs [23,24]. This is a large region in the genome with a LD spanning around 1.6 megabases without any recombination. The haplotype diversity is caused by the 970 kb inversion polymorphism, H1 has direct orientation and H2 has an inverted orientation [25,26]. H1 haplotypes are associated with AD, amyotrophic lateral sclerosis, PD and other neurodegenerative diseases [25]. H1 haplotype occurs in all populations with reasonable recombination and variability. H2 haplotype is of European origin and shows very limited variation in sequence [25,26]. This inversion contains many genes like MAPT, CRHR1, NSF, KANSL1, LRRC37A, LRRC37A2, ARL17A, ARL17B [27]. While the associations of MAPT haplotypes and neurodegenerative diseases are well established, the functional mechanism behind these associations is not clear.

The purpose of current study was to analyse the effects of the known variants at the ApoE, MAPT and SNCA loci (rs356181, rs3910105) on transcriptional regulation. We utilised data from the Parkinson’s Progression Markers Initiative (PPMI) project where the genotypes of 570 subjects and their blood transcriptome data are available. We aimed to identify what transcripts or genes are differentially regulated by the genetic variants at these loci. As a result, we identified profound regulatory effect of ApoE haplotypes on TOMM40 expression and differential transcriptional regulation by the H1 or H2 haplotypes at the MAPT locus. In addition to the *cis* regulatory loci, we identified many *trans* locations that are also controlled by the variations in these haplotypes.

## 2. Materials and Methods

### 2.1. Datasets

In this study we utilized the Parkinson’s Progression Markers Initiative (PPMI) cohort data that were downloaded from www.ppmi-info.org/data (accessed on 19 January 2021). The PPMI is a longitudinal cohort to follow Parkinson’s patients and to describe the course of Parkinon’s disease. The dataset contains whole transcriptome data from the blood together with genetic and clinical data. The clinical dataset already contained genotypes or haplotypes ApoE, Apo e4, MAPT, rs356181 and rs356181 loci. Genotyping data available for the PPMI cohort are based on the NeuroX SNP chip results. ApoE e2, e3 and e4 haplotypes were defined by the rs429358 CT and rs7412 CT genotypes. TT is for e2, TC is for e3 and CC is the e4 haplotype. MAPT haplotypes were defined by rs17652121, rs8070723, rs1052587, rs16940799 and rs17652748 genotypes where minor alleles for each position were counted and H1H1, H1H2 and H2H2 were detected. rs356181 and rs356181 are based on genotyping chip results. 

Whole blood RNAseq data were downloaded from the PPMI website and transcript-based annotation was used for further analysis. This is the release of Phase 1 and Phase 2 PPMI RNA-seq data that were already processed, mapped to reference genome hg 19 and with the counts data generated. Briefly, FASTQ files were mapped to hg19 (GRCh37) by STAR on GENCODE v19, counts were created for genes and transcripts using FeatureCounts and abundance estimates (transcripts per Million, TPM) via Salmon. All this prerequisite work was done as a part of ongoing PPMI RNA-seq analysis and was conducted by HudsonAlpha Institute for Biotechnology, Institute of Translational Genomics of the University of Southern California and The Translational Genomics Research Institute, TGen. As the RNAseq data were annotated with the hg19 version of human genome all other annotations were also based on hg19. In this analysis all subjects, PD (314) and SWEDD (49) cases and controls (156), were combined and transcript expression signals were tabulated after importing Salmon files in the R to prepare them for the eQTL analysis. Altogether, 95,309 transcripts were used for the analysis in combination with the five genetic loci APOE, APOE_e4, MAPT, SNCA rs356181 and SNCA rs3910105.

### 2.2. eQTL Analysis

Matrix eQTL was used to calculate the genetic loci regulating the expression transcript variants [28]. We used additive linear model with covariates, age and sex, with FDR threshold 0.05. During eQTL analysis, local (*cis*) and distant (*trans*) quantitative loci were called, distant locus threshold was set on 1M bp. Raw results were used for *circos* plotting and plotting using R *ggbio* and *ggpubr* packages [29,30]. 

The correction for multiple testing of eQTL was performed using FDR and only results that remained significant after FDR correction are reported here. For pairwise comparisons between the genotype Wilcoxon test was used and *p*-values were challenged with the Bonferroni multiple comparison test.

## 3. Results

### 3.1. General Analysis for All Markers

Matrix eQTL analysis identified 151 eQTL loci, 29 in *cis* and 122 in *trans* position as statistically significant with the FDR below 0.05 (Appendix A). This ranged from largest effect on gene expression was β value 256 of MAPT haplotype on the LRRC37A2-201 transcript. The second largest was MAPT effect on KANSL1-002 transcript (β 153). The highest down-regulating effect on gene expression was again from MAPT haplotypes on the expression of LRRC37A2-001 (β value −178). MAPT haplotypes had strong negative effects on the KANSL1-006 transcript, LRRC37A2-201 transcript and on other transcripts in the MAPT locus. We identified transcript specific regulatory effect in all studied loci. Overall summary of the identified eQTLs is given in Table 1 and detailed information is in Appendix A. Appendix A also contains the exact Ensembl codes for the transcript variants. We built a circos plot based on Appendix A to show all the combined identified effects (Figure 1). More detailed circus plot is available as Appendix A.

ApoE locus was analysed by using all three haplotypes and their combinations separately and only addressing the effect of e4 haplotype. ApoE e4 alone gave much larger number of eQTL targets than the separate e2, e3, e4 ApoE haplotypes combined demonstrating the overwhelming effect of ApoE e4 haplotype (Figure 1, “APOE” and “APOE_e4”). ApoE demonstrated more suppressing or downregulating effects that ApoE e4, however most of the identified eQTLs had stimulating effects on the target transcripts. Six different combinations of the ApoE haplotypes possessed strong and statistically significant effect on the expression of the TOMM40 gene (Figure 2). In Figure 2 we show the effect on the TOMM40 gene where the expression signals from different transcripts were combined. ApoE e3e3 had the strongest expression signal followed by e2e3 haplotype. The lowest expression was caused by e4e4 haplotype and any haplotype combination with at least single e4 genotype showed very low expression level of TOMM40 compared to any other haplotype combinations. We identified that ApoE e3 allele is profoundly stimulating for TOMM40 expression, whereas e4 is significantly suppressive. The e2 allele also seems to have repressive effects but not as significant as the e4 variant. Our analysis showed that the main effect of the ApoE haplotypes comes from the stimulating effect of the e3 allele on TOMM40 expression. 

SNCA rs3910105 had a dominating repressive effect to its targets that is evident from Figure 1 and Table 1. This feature is quite unique compared to other variants used in the present study.

### 3.2. Analysis of the Individual Markers on Transcripts

#### 3.2.1. All ApoE Haplotypes

As described in previous section, ApoE haplotypes possessed a strong stimulating activity on the TOMM40 gene and this effect was caused by the e3 variant. Haplotype e3e3 had the highest level of expression and any haplotype with single e3 (e2e3 and e3e4) were the next highest expressing variants. Here, we analyzed the effect of these haplotypes to the transcript level expression profile (Figure 3). Indeed, like with the gene level TOMM40 expression, e3 again showed the strongest stimulating effect on two TOMM40 transcripts, TOMM40-001 and TOMM40-005 (Figure 3). Transcript level analysis showed that while combination e3e4 had significantly downregulated expression of both isoforms, e2e3 had similar expression with the e3e3 haplotype. This is in contrast with the gene level analysis shown at the Figure 2 where e2e3 had a statistically significant reduction in expression compared to e3e3. Transcript based analysis shows that e2 variant may possess some transcript specific stimulating effect. ApoE e4 variant had the lowest expression level for transcripts.

#### 3.2.2. ApoE e4 Effect

As ApoE e4 had a strong suppressive effect on combined TOMM40 gene expression (Figure 2), we next performed the analysis using individuals grouped by the number of e4 alleles they possessed. There were three combined haplotypes; those with no e4 alleles, with one e4 or two e4 alleles. As a result, we identified a robust suppressive effect of e4 on the expression of all TOMM40 transcripts that were significantly affected in the matrix eQTL analysis (Figure 4). 

Interestingly, ApoE e4 also had a significant repressive effect on the expression of sarcosine dehydrogenase transcript SARDH-003 (Appendix A). The SARDH gene encodes an enzyme that localizes in the mitochondrial matrix and is responsible for oxidative demethylation of sarcosine. It is remarkable that ApoE e4 had very selective repressive effect only to four transcripts of two genes that all are involved in mitochondrial function. TOMM40 has seven transcript variants from which six are protein coding and SARDH gene has eight transcript variants with seven of them being protein coding. All other ApoE e4 effects were activating for their target genes. Therefore, ApoE e4 genetic effect might be caused by its involvement in regulating the transcripts of TOMM40 and SARDH genes.

#### 3.2.3. MAPT Haplotypes

Genetic variation at the MAPT locus altogether affected 47 genomic regions with 18 *cis* and 29 *trans* effects. Transcripts of the genes LRRC37A2, KANSL1, ARL17B, LRRC37A, ARHGAP27, PLEKHM1, MAPT, CRHR1, ARL17A were affected by the MAPT haplotype by *cis* regulation. All these genes were from the locus 17q21.31. Moreover, four transcripts of the gene LRRC37A3 in the 17q24.1 were also significantly affected (Appendix A). The β values of the model ranged from −178 to 256 indicating a robust regulatory effect on the genes. In many cases different transcripts of the same gene had opposite effects in a sense that some transcripts were up-regulated and the others were downregulated. For instance, the transcript CRHR1-202 was upregulated by the H2H2 haplotype, while the CRHR1-201 was downregulated by the H2H2. On the other hand, a single transcript of ARL17A was up-regulated by the H2H2 haplotype (β 2.3) and two transcripts of the ARL17B were both downregulated by H2H2 (β values −59 and −19).

The effects of the MAPT haplotypes on expression were the strongest of all the variants analyzed in this study. The largest upregulation by H2H2 haplotype was identified for the transcripts LRRC37A2-201, KANSL1-002 and LRRC37A-003 (Figure 5) with H2 acting in a dose—dependent manner. 

The most significant repressive effect of MAPT haplotypes was also evident for the locus 17q21.31. The most significant and the largest downregulation by H2H2 haplotype was identified for the transcripts LRRC37A2-001, KANSL1-006 and LRRC37A-002 (Figure 6). These are different transcripts but from the same genes that had the strongest up-regulating effect. This finding indicates MAPT haplotypes have a bidirectional effect that is transcript isoform specific indicating a molecular specificity of the H2H2 haplotype in regulating the locus. The single transcript from MAPT gene itself was downregulated by the H2H2 haplotype. Two transcripts from another promising target for neurodegenerative disease, PLEKHM1, were both down-regulated by the H2H2 haplotype.

#### 3.2.4. SNCA Locus rs356181

The SNP rs356181 is an intronic variant in the gene LOC105377329, that is in close proximity to SNCA with potential association of its expression. We identified that variation at rs356181 modified expression of SNCA transcripts. SNCA-007 is downregulated by the CT and TT genotypes, whereas SNCA-008 is upregulated by CT and TT genotypes (Figure 7). 

#### 3.2.5. SNCA Locus rs3910105

SNP rs3910105 is an intronic variant in the SNCA gene. This SNP regulated nine different targets, four were in *cis* location. Three targets were different SNCA transcripts and the fourth gene was MMRN1, adjacent to SNCA (Figure 8). Our analysis indicated that CC, CT and TT genotypes differentially regulated these transcripts. Both MMRN1 and SNCA-008 were downregulated by CT and TT genotypes. However, SNCA-007 and SNCA-009 were significantly upregulated by CT and TT variants in the rs3910105. In regard to the SNCA gene transcripts, the regulatory pattern is reverse compared to rs356181 that indicates specific regulatory pattern of these SNPs. 

## 4. Discussion

In this study we describe the eQTL analysis performed using PPMI blood RNAseq data and genotypes for ApoE, MAPT, rs3910105 and rs356181. We identified that these genetic loci possess highly significant regulatory functions for at least 151 transcripts across the genome. Our results highlight new findings that may explain in part the functional consequences of these well-defined genetic variants in neurodegenerative diseases.

We identified that the TOMM40 gene expression (Figure 2) and transcript levels (Figure 3 and Figure 4) are dependent on the ApoE haplotypes. Subjects with ApoE e3e3 haplotypes had the highest level of TOMM40 expression and even those with a single e3 allele had significantly increased TOMM40 expression in the blood. This is the first time that a connection between the ApoE haplotypes and TOMM40 gene and transcript expression has been established. The genetic link between ApoE and TOMM40 has been well established [9,10,11,31,32,33]. Most of these studies have looked for genetic interaction between ApoE and TOMM40 polyT polymorphisms to explain the differences in the AD sub-phenotypes and in the ages of the disease onset [10]. There are few studies where this genetic interaction is analysed in the context of transcriptional regulation and cis-regulatory effect caused by polyT polymorphism has been described [34]. In that sense, we are not the first to describe functional regulatory interaction between TOMM40 and ApoE, but our finding links ApoE haplotypes to the levels of the TOMM40 gene and transcripts. More precisely, we identified the e3 haplotype as a high-expression haplotype and e4 as low expression haplotype. We also identified that e4 haplotype regulates expression of another mitochondrial gene, SARDH. Our results connect ApoE haplotypes directly to the function of mitochondria and help to explain the neurodegeneration caused by the ApoE variants as the impact of mitochondrial dysfunction in neurodegeneration is well-established [35]. 

The model for mitochondrial mechanisms and changes in mitochondrial functions underpinning the ApoE effects was proposed several years ago [31]. Recent genetic studies have also indicated that ApoE haplotypes are also related to PD and other neurodegenerative conditions and are responsible for the cognitive decline in these patients [8]. The role of TOMM40 in mitochondrial dysfunction is further supported by existing literature where a reduction in protein expression levels has been found in neurodegenerative diseases [36]. All these findings are consistent with the genetic effect of ApoE haplotypes being mediated by its ability to regulate the expression of TOMM40 and to induce (or avoid) mitochondrial dysfunction. Protective e3e3 haplotype of ApoE induced significant expression of TOMM40 and its transcripts. This expression difference could be the mechanism behind the protective effect of ApoE e3e3.

We cannot exclude the possibility that the e3e3 effect is caused by the unbalanced distribution of the e3 allele as it is the most abundant version of the ApoE haplotypes. This is unlikely the case, as we also see the significant differences with heterozygous haplotypes e2e3 and e3e4 where the numbers of the subjects are larger. However, authors feel that extra experimental work is needed to get biochemical evidence for this association.

MAPT haplotypes H1H1, H1H2 and H2H2 are possibly the most studied haplotypes in neurodegenerative disease research. While this haplotype and locus is interesting as it hosts the MAPT gene that encodes tau protein, the exact molecular mechanism why the MAPT locus is associated with neurodegenerative disease is not understood [27]. Our report identified new potential candidates that are regulated by the variants at this locus. The haplotypic variation induced statistically significant changes in expression of many genes at this locus. Affected transcripts were from LRRC37A2, KANSL1, ARL17B, LRRC37A, ARHGAP27, PLEKHM1, MAPT, CRHR1 and ARL17A genes that reside in the locus 17q21.31 and also from the LRRC37A3 gene that is in the adjacent locus 17q24.1. Transcriptional differences were bidirectional in many cases. Some transcripts of LRRC37A2, KANSL1 and LRRC37A were upregulated whereas others were downregulated. This specificity indicates functional impact behind eQTL analysis. Additional support for real functional regulatory activity comes from the size of the effect that we measured with β values. The largest (up to 256) and the lowest β values (−178) were related to the regulation of aforementioned genes and transcripts. This large variation in β values indicates distinct regulatory effect that H1 or H2 genotypes on specific targets. The genes that are affected by this haplotypic variation are all functional candidates for the pathogenetic model of neurodegeneration. KANSL1 and MAPT are primary candidates, but PLEKHM1 is often a neglected one. PLEKHM1 is a well characterized protein that is involved in the endosomal system and autophagy and is necessary for autophagosome-lysosome fusion [37]. Endosomal pathway and autophagy are both pathways that are severely affected during neurodegeneration making PLEKHM1 an interesting candidate for further studies.

Our analysis also involved two SNPs from the locus of SNCA and we identified several targets regulated by these variants. Both SNPs affected the differential expression of SNCA transcripts and rs3910105 also affected the MMRN1 gene that is adjacent to the SNCA gene. MMRN1 is elastin microfibril interface 4 and is able to carry platelet factor V. A recent study identified that MMRN1 expression was increased in SNCA knockout mice that was attributed to the genetic modification in the SNCA locus [38]. However, our study identified regulatory link between the intronic SNP in SNCA gene and MMRN1 gene expression. Additional experimental studies are needed to identify the nature of this interaction.

Strengths of our study include the large study cohort consisting of data from 570 individuals and the DNA and RNA were from the same tissue, blood. As we had both RNA and DNA from the same persons and from identical tissues our study had minimal confounding factors. During the linear modelling we adjusted for the age and sex of the participants. Using the PPMI dataset has additional advantage as the research community can access cell lines from some of these individuals for functional verification experiments. 

## 5. Conclusions

In conclusion, we identified new target genes that are regulated by the common variation relevant to the risk of neurodegenerative diseases. ApoE haplotypes significantly regulate transcription of the TOMM40 gene and the e3 variant strongly upregulates expression of TOMM40, whereas e4 downregulates its expression. This regulatory connection with TOMM40 can explain the genetic effects of ApoE that has been described decades ago.

MAPT H2H2 haplotype is associated with profound differential effect on the transcripts of the genes that have functional impact on neurodegeneration. This differential expressional control over KANSL1, LRRC37A2 and LRRC37A transcripts has very large biological effect and therefore may explain the molecular mechanism for the MAPT haplotypes in neurodegenerative processes.

## Figures and Tables

**Figure 1 genes-12-00423-f001:**
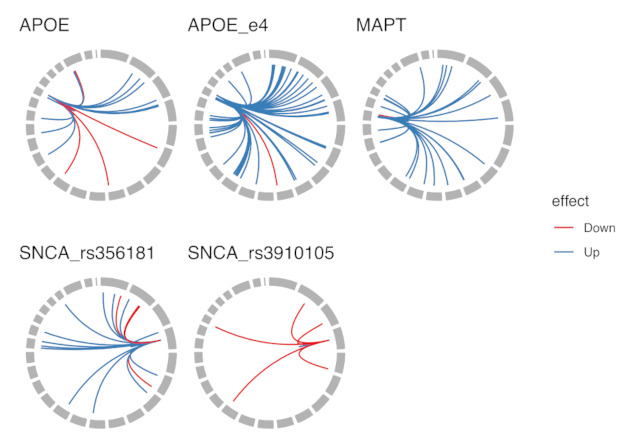
Combined Circos plot describing the regulatory effects of all studied loci. Each locus is shown in a separate panel with *cis* and *trans* activities combined into single circus plot. Down or up-regulating effects are shown with red and blue colour, respectively. Some loci had dominatingly up-regulating effects, while the one locus had mainly down-regulating effects.

**Figure 2 genes-12-00423-f002:**
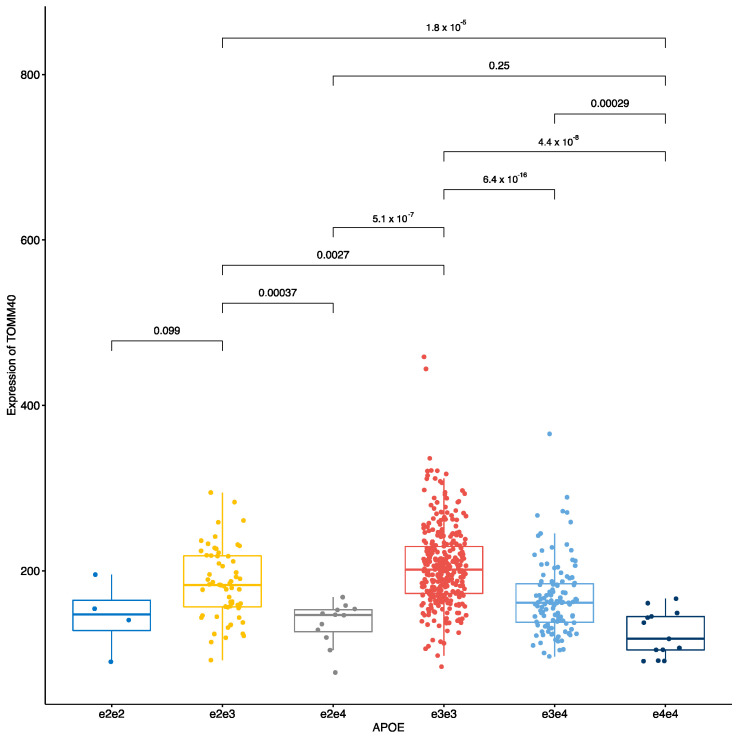
ApoE haplotypes differentially regulate the expression of the TOMM40 gene. The gene signal in this figure is aggregated from all RNA isoforms. ApoE e4 haplotype has statistically significantly lower expression than other haplotypes. Even single e4 allele was sufficient to significantly reduce expression of the TOMM40 gene. The bars in this figure and in all other figures show pairwise comparisons and respective *p*-values after the Wilcoxon test.

**Figure 3 genes-12-00423-f003:**
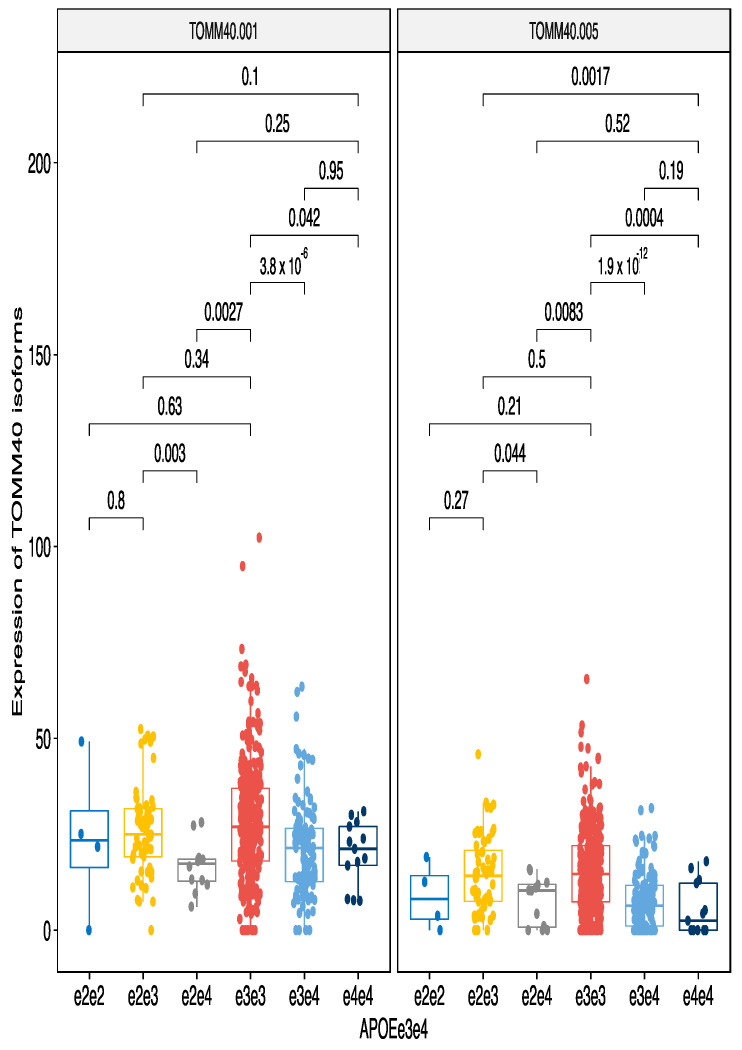
ApoE haplotypes regulate expression of the TOMM40 transcripts 001 and 005. ApoE e4 haplotype has statistically significantly lower expression than other haplotypes. Even a single e4 allele was sufficient to significantly reduce expression of the TOMM40 isoforms. The bars show pairwise comparisons and respective *p*-values after the Wilcoxon test.

**Figure 4 genes-12-00423-f004:**
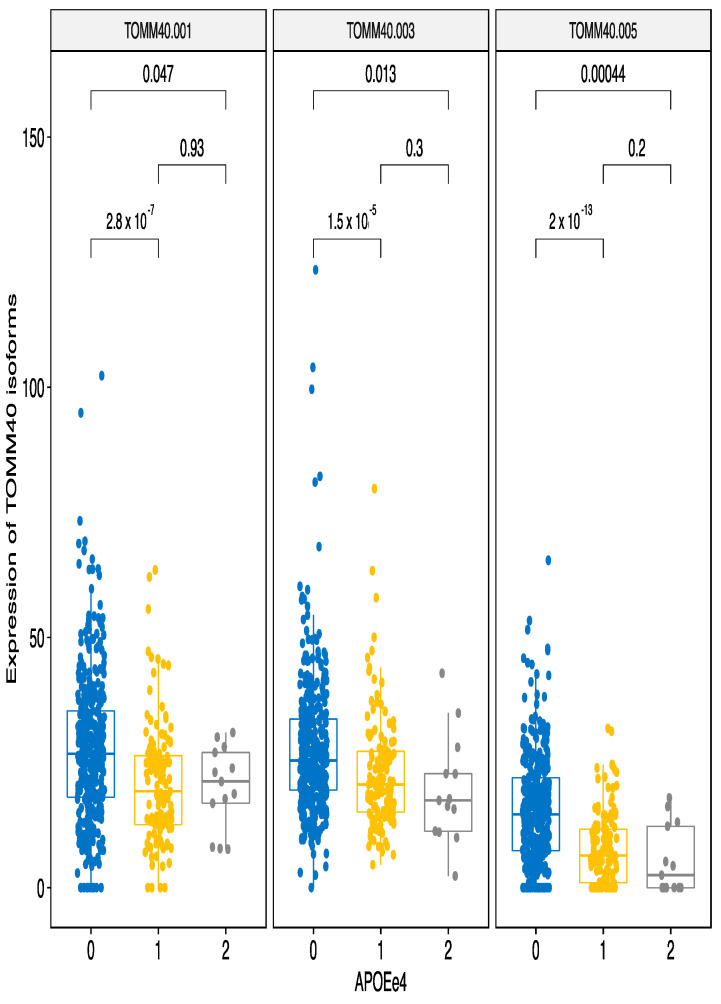
ApoE e4 haplotypes down-regulate expression of the TOMM40 transcripts 001, 003 and 005. Even a single e4 allele is sufficient to significantly reduce expression of the TOMM40 isoforms. The bars indicate pairwise comparisons and respective *p*-values after the Wilcoxon test.

**Figure 5 genes-12-00423-f005:**
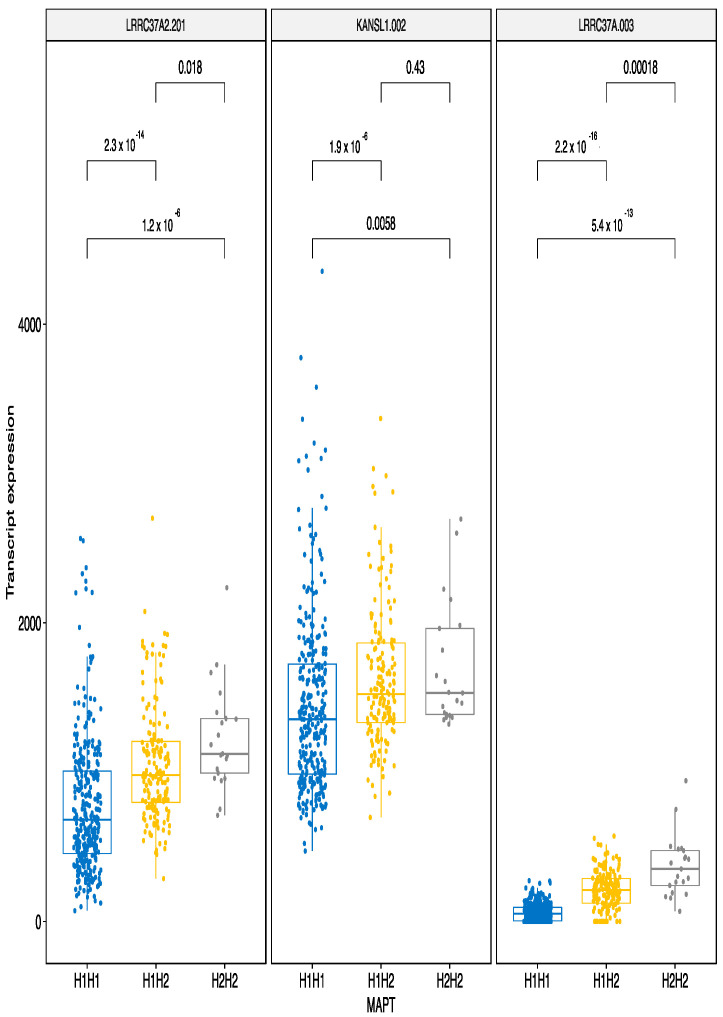
MAPT H2H2 haplotype upregulates several transcripts, three with the largest β values are illustrated here. Even a single H2 variant is sufficient to increase the expression of given transcripts. The bars indicate pairwise comparisons and respective *p*-values after the Wilcoxon test.

**Figure 6 genes-12-00423-f006:**
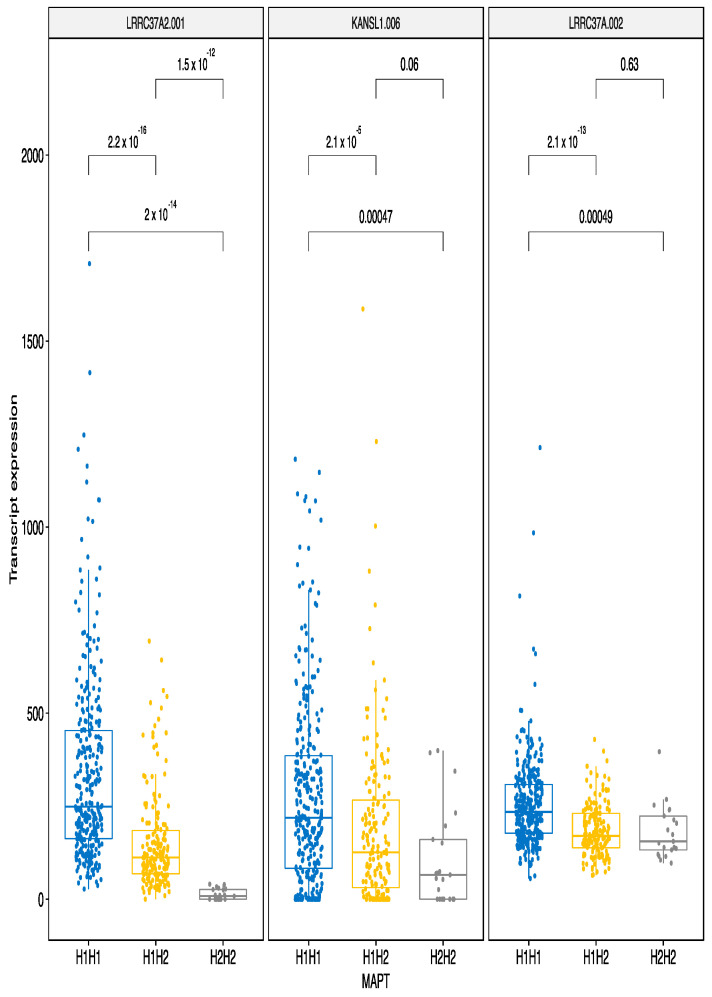
MAPT H2H2 haplotype downregulates several transcripts. LRRC37A2, KANLS1 and LRRC37A transcripts had the highest negative β values (down-regulating effect) and are illustrated here. These are different transcripts from the same genes that are shown in Figure 5 and were up-regulated by the same haplotype. Even a single H2 variant is sufficient to downregulate the expression of given transcripts. The bars indicate pairwise comparisons and respective *p*-values after the Wilcoxon test.

**Figure 7 genes-12-00423-f007:**
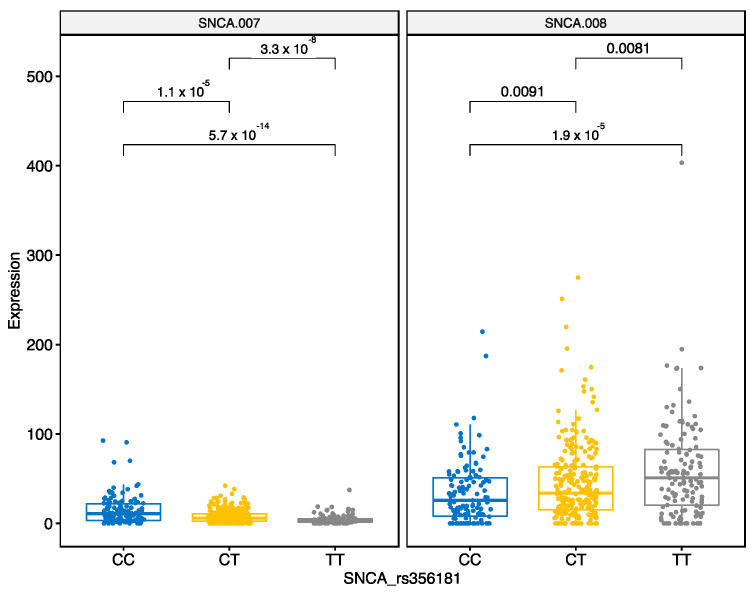
Different genotypes of the SNP rs356181 (CC, CT or TT) differentially regulate expression of two SNCA transcripts. The bars indicate pairwise comparisons between CC, CT and TT genotypes and respective *p*-values after the Wilcoxon test.

**Figure 8 genes-12-00423-f008:**
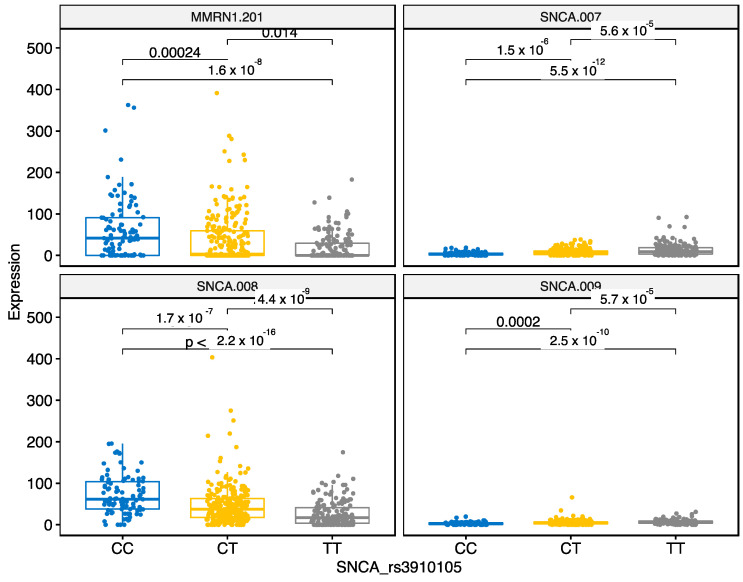
Different genotypes of the SNP rs35910105 (CC, CT or TT variants) differentially regulate expression of three SNCA transcripts and MMRN1. The bars indicate pairwise comparisons and respective *p*-values after the Wilcoxon test.

**Table 1 genes-12-00423-t001:** The overview of the eQTL loci, number of transcripts and their effect sizes as β values for blood transcriptome. Explanation: Variants—variants under the study, N—number of statistically significant transcripts identified for particular variant, Min β—minimal β value, Max β—maximum β value, Mean β—mean β value, SD—standard deviation, SE—standard error of mean.

Variants	N	Min β	Max β	Mean β	SD	SE
ApoE	18	−18.7	0.7	−1.9	4.9	1.2
ApoE e4	58	−6.7	15.9	1.0	3.3	0.4
MAPT	47	−178.5	256.6	1.9	60.8	8.9
rs356181	19	−20.5	52.6	3.9	14.6	3.4
rs3910105	9	−21.2	5.3	−4.1	9.7	3.2

## Data Availability

Raw data are available from the PPMI website (www.ppmi-info.org/data, accessed on 19 January 2021).

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
