# Peer review of "Transcript Variants of Genes Involved in Neurodegeneration Are Differentially Regulated by the APOE and MAPT Haplotypes"

_genes, 2021, doi:10.3390/genes12030423_

Round 1
Reviewer 1 Report
In principle a solid and interesting study. However, the methods and datasets are not described well and the reader needs to go forward and backwards many times
to fully understand the study, the applied methods and models. That in particular would need strong improvements.
Comments:
----------------
Of course it is a matter of taste, but I was under the impression that the alleles e2,e3,e4 are normally donated with an epsilon instead of 'e'?
Is there any particular reason you used the assembly hg19 and not e.g. hg38?
Chapter2: It is not clear to me, have you downloaded FASTQ files and based on that quantified the gene expressions by yourself or have you downloaded
from PPMI the genotypes and gene expressions already ready processed? Please elaborate this in more detail, also the quantification pipeline should be
described (tools, filtering, trimming, etc.). This is especially important, as you you isoform level quantification and I would like to know, how the
reads were assigned to different transcripts of the same gene.
Also, what kind of expression values do you use? TPM, FPKM?
Chapter2.2: I miss here some more details on the data, how many different genes have you the expression data and how many loci did you use in the eQTL
analysis?
Chapter 3.1: Please describe you used statistic 'Beta' in the M&M part in more detail.
l.88: Could you explain in more detailyour variants, what is ApoE? Is it the e2,e3 or e4 variant? This is not very clear described, what you actually
use and how you donate it. One supplement table with variant name, location and REF and Alt allele would do. E.g. what I do not understand, e.g.
for the ApoE variant, what is your eQTL model? I thought you would use all haplotype permutations, but then the model is getting rather complex?!
l.98: Could you explain what is SWEDD?
l.106: Please cite those R packages
l.126: Do you mean 'combined' instead of 'separately'?
l.157: Please discuss if the effect of e3 could also due to chance, apparently most of your observations have e3e3, so could it be that this is just by chance
and due to the small sample size in other haplotype groups?
Figure1: Also a matter of taste, but wouldn't it make the figure more space effective to have two rows and three columns?
Figure1: Please indice the location of the variant on the circle. It would increaase the readability.
Figure1: Is there the cis-panel missing? I have the feeling I am looking only at trans-eQTLs there?!
Figure2 (and others): How have you tested for different expressions?! It is nowhere described, as far as I can see it. Also, you would need to correct
also there for multiple testing, wouldn't you?
Figure8: What are the mean expression values of SNCA.009? Is it sufficiently high expressed to do any meaningful analysis?
Reviewer 2 Report
===========
Genes
-----------------
Manuscript:
“Transcript variants of genes involved in neurodegeneration are differentially regulated by the APOE and MAPT haplotypes,” by Sulev Koks, Abigail Pfaff, Vivien Bubb, and John Quinn.
-----------------------------------------------------------------
The manuscript presents the results of statistical analyses of the effects of genetic variants from the APOE, MAPT, and SNCA gene loci on transcriptional regulation.
The authors identified links between the APOE polymorphism and the TOMM40 gene expression, between MAPT haplotypes and expression of several genes in the same genetic region. These new results are of high interest because they highlight links between different genes involved in neurodegenerative processes, which can be helpful for improving our understanding of mechanisms underlying AD and PD development and progression.
I have a few comments that I hope can help to make the manuscript stronger.
(1) The authors do not provide confidence intervals, SEs, or SDs through the manuscript (see Table 1S), neither they describe them in Figs.2-8. Therefore, it is difficult to make conclusions about differences in expression levels related to different APOE alleles and MAPT haplotypes.
(2) There are no explanations/notes about abbreviations, column names etc in Tables and Figures.
In Supplementary Table 1S, the column names and meaning of the numbers and abbreviations/text are not described.
In Supplementary Table 1S, it would be helpful to have chromosome numbers for each locus and confidence intervals, SEs or SDs.
Figures 2-8, there are no descriptions about the meaning of vertical lines in the boxplots.
(3) In Fig 1S, there is no any explanation for this figure, which makes it difficult to understand and interpret.
(4) In Figs. 1 and 1S, the same things (down- and up-regulation) are represented by different colors. I would suggest harmonizing the use of colors within the manuscript, i.e. the same color meaning in different figures.
(5) Several times the authors used a phrase “the smallest effect” to emphasize the largest down-regulating effect. Usually, the smallest effect refers to an effect with a smallest magnitude (close to zero), but not to the negative effects with a large magnitude.
(6) It would be helpful if the authors can provide information about the sample(s) used in the analyses (sample size, biodemographic information, composition etc).
(6) In the “Materials and Methods” section there is insufficient description of statistical methods and models used. For instance, a short description about the methods of calculating p-values in Table 1S would be helpful. Moreover, it is unclear how p-values in Fig.2 (and other figures) were obtained. Corresponding procedures were not described in the “Materials and Methods” section.
(7) Can the authors explain how they calculated p-values displayed in Figs.2-8. In these figures, for some significant p-values the quartile levels intersect, which can refer to non-significant differences. For instance, Figure 2 provides no evidence of statistically significant differences between the associations of the APOE e4e4 genotype with respect to other genotypes.
(8) In "Discussion", the authors wrote: “We identified that the TOMM40 gene and transcript levels are dependent on the ApoE haplotypes.” It is unclear what does it mean that the TOMM40 gene is dependent on the ApoE haplotypes. Can the authors provide more details?
(9) In the same paragraph (Discussion, paragraph 1) the authors wrote: “Those with ApoE e3e3 haplotypes had the highest level of TOMM40 expression and even those with a single e3 allele had significantly increased TOMM40 expression in the blood.” If the ApoE e3e3 haplotype leads to the highest level of TOMM40 expression how a single e3 allele leads to significantly increased TOMM40 expression in the blood? Does it mean that the authors know a reference level of TOMM40 expression? Is this level defined by the APOE e2 or e4 allele? Should this level have the same value in all tissues?
(10) In the Discussion section (lines 297-299) the authors state that “Protective e3e3 haplotype of ApoE induced significant expression of TOMM40 and its transcripts. This expression difference could be the mechanism behind the protective effect of ApoE e3e3.” From this statement one can conclude that the lower the TOMM40 expression level the lower risk(s) of neurodegenerative disorder(s) in respective group(s) of individuals. Can the authors explain how the lower TOMM40 expression (see Figs.2,3) is related to the decreased risks of AD in the APOE e2 allele carriers?
(11) Fig.1 is not informative. I would suggest improving it by showing chromosomes and a few genes related to the manuscripts. The caption to this figure should be substantially revisited by providing additional information, which can facilitate understanding of the presented results.
(12) In the MAPT gene region, there is a set of SNPs belonging to other genes in this region. Those SNPs are in LD to the SNPs that define MAPT haplotypes. Could the authors provide an explanation how it can be distinguished that the considered MAPT SNPs, but not their proxies from other genes, affect the expression levels?
Round 2
Reviewer 1 Report
I think the manuscript improved greatly! While reading I spotted one small typo in line 121 "results"